# Human Erbb2-induced Erk activity robustly stimulates cycling and functional remodeling of rat and human cardiomyocytes

Nicholas Strash[1], Sophia DeLuca[1], Geovanni L Janer Carattini[2], Soon Chul Heo[2], Ryne Gorsuch[2], Nenad Bursac[1,2]*

[1]Department of Cell Biology, Duke University, Durham, United States; [2]Department of Biomedical Engineering, Duke University, Durham, United States

**Abstract** Multiple mitogenic pathways capable of promoting mammalian cardiomyocyte (CM) proliferation have been identified as potential candidates for functional heart repair following myocardial infarction. However, it is unclear whether the effects of these mitogens are species-specific and how they directly compare in the same cardiac setting. Here, we examined how CM-specific lentiviral expression of various candidate mitogens affects human induced pluripotent stem cell-derived CMs (hiPSC-CMs) and neonatal rat ventricular myocytes (NRVMs) in vitro. In 2D-cultured CMs from both species, and in highly mature 3D-engineered cardiac tissues generated from NRVMs, a constitutively active mutant form of the human gene Erbb2 (cahErbb2) was the most potent tested mitogen. Persistent expression of cahErbb2 induced CM proliferation, sarcomere loss, and remodeling of tissue structure and function, which were attenuated by small molecule inhibitors of Erk signaling. These results suggest transient activation of Erbb2/Erk axis in CMs as a potential strategy for regenerative heart repair.

**\*For correspondence:**
nenad.bursac@duke.edu

**Competing interest:** The authors declare that no competing interests exist.

## Introduction

The adult mammalian heart is composed primarily of post-mitotic cardiomyocytes (CMs) (*Bergmann et al., 2009*; *Mollova et al., 2013*). Due to an apparent lack of resident stem cells (*van Berlo et al., 2014*), the heart is unable to replace lost CMs following myocardial infarction (MI), and instead undergoes fibroblast-mediated scarring, resulting in a decline in cardiac function. One potential approach to regenerate the heart following MI is to stimulate the proliferation of endogenous CMs by gene therapy targeting cell cycle regulation (*Leach and Martin, 2018*; *Lin et al., 2014*; *Mohamed et al., 2018*; *Tzahor and Poss, 2017*). While inducing adult CM proliferation is only the first step toward achieving efficient cardiac repair post-MI, it is arguably the most challenging and important one. A major question in the field, however, is that gene or pathway activation in particular is optimal for inducing human CM proliferation and restoration of function.

Several approaches involving increased expression of cell cycle-related proteins and modulation of the Hippo, neuregulin (Nrg1), or Wnt signaling pathways have been shown to generate robust proliferative responses in zebrafish and rodent CMs. Specifically, overexpression of Cyclin D2 alone or a combination of Cyclin-dependent kinase 1 (CDK1), CDK4, Cyclin B1, and Cyclin D1 promoted cell cycle activation in post-mitotic mouse, rat, and human CMs in vitro and in mouse hearts in vivo (*Fan et al., 2019*; *Mohamed et al., 2018*; *Zhu et al., 2018*). Hippo pathway modulation via overexpression of constitutively active (ca) Yap variants (incapable of being phosphorylated and degraded) led to robust proliferative responses in vitro and in vivo through chromatin alteration surrounding

proliferation-inducing genes (*Byun et al., 2019*; *Monroe et al., 2019*). The mitogen Nrg1 induced CM proliferation in zebrafish through activation of its co-receptor, Erbb2 (*Bersell et al., 2009*; *Gemberling et al., 2015*), and overexpression of caErbb2 promoted CM proliferation in adult mice by inducing CM dedifferentiation (*D'Uva et al., 2015*), possibly via Yap activation (*Aharonov et al., 2020*). Wnt stimulation by small-molecule inhibition of GSK3β (*Buikema et al., 2020*; *Fan et al., 2018*; *Mills et al., 2019*) or induction of β-catenin release from the cell membrane (*Fan et al., 2018*) induced proliferation in human CMs in vitro. Yet, there have been no studies that directly compare the pro-proliferative capacity of these different mitogenic pathways in parallel.

In this report, we directly compared the effects of manipulating different mitogenic pathways on in vitro cell cycle activation in CMs from various species and maturation levels, including 2D mono-layer cultures of human-induced pluripotent stem cell-derived CMs (hiPSC-CMs; *Shadrin et al., 2016*; *Zhang et al., 2013*) and neonatal rat ventricular myocytes (NRVMs), as well as functional 3D engineered NRVM tissues (cardiobundles) (*Helfer and Bursac, 2020*; *Jackman et al., 2016*). We then probed the mechanisms underlying the observed mitogenic responses. We found that the human, but not rat (*D'Uva et al., 2015*), caErbb2 ortholog (cahErbb2) induced the most robust proliferative effects in hiPSC-CMs and NRVMs, which were associated with significant changes in CM morphology and function and mediated via upregulation of the Erk signaling pathway.

## Results

### Lentiviral expression of mitogens promotes hiPSC-CM proliferation without inducing apoptosis

Lentiviral vectors (LVs) were designed with the MHCK7 promoter (*Salva et al., 2007*) driving muscle-specific expression of mitogens together with a mCherry reporter (*Figure 1—figure supplement 1*) and used to transduce hiPSC-CMs and NRVMs. We applied a flow cytometry strategy (*Figure 1—figure supplement 2*) validated using an established hiPSC-CM mitogen (GSK3 inhibitor, CHIR99021) (*Buikema et al., 2020*; *Mills et al., 2019*) to determine whether an LV-induced mitogen expression resulted in cell cycle activation or apoptosis specifically in mCherry-labeled CMs. Compared to LV-driven expression of mCherry only, hiPSC-CMs transduced with LVs (*Figure 1A*) encoding caCtnnb1 (ca β-catenin), Ccnd2 (Cyclin D2), carErbb2, or cahErbb2, but not caYap8SA, exhibited significantly higher incorporation of 5-ethynyl-2′-deoxyuridine (EdU), indicating greater DNA synthesis (*Figure 1B*). Increased numbers of CMs expressing phosphorylated histone H3 (H3P), an indicator of the mitotic (M) phase of the cell cycle, were also found with cahErbb2, caCtnnb1, and Ccnd2 transduction, with cahErbb2 appearing to have the most robust effects (*Figure 1C*). Since cultured hiPSC-CMs are relatively immature, we also assessed whether cahErbb2 treatment would increase cell cycle activity in hiPSC-CMs cultured in a maturation medium (MM) shown to reduce proliferation and promote contractile function in human CMs (*Mills et al., 2017*). While the MM reduced EdU incorporation in CMs compared to our standard culture medium, cahErbb2 LV treatment in MM consistently increased EdU incorporation and H3P expression in CMs (*Figure 1—figure supplement 3*). Furthermore, we observed comparable percentages of diploid and polyploid hiPSC-CMs between control and LV-treated total or EdU$^+$ CMs (*Figure 1D and E*). Collectively, these results showed that hiPSC-CMs with LV-mediated expression of cahErbb2, caCtnnb1, and Ccnd2 were induced to enter DNA synthesis and mitotic phases of the cell cycle at a higher rate than control hiPSC-CMs, and continued to undergo successful cytokinesis. Additionally, from cleaved caspase-3 (Cc3) analysis, transduced mitogens did not increase CM apoptosis, with caYap8SA having an anti-apoptotic effect (*Figure 1F*). In contrast, both increased DNA synthesis and mitosis as well as pro-apoptotic effects were found with the application of CHIR99021 (*Figure 1—figure supplement 2B*; *Mills et al., 2017*). Taken together, increased EdU incorporation with no change in CM polyploidy or apoptosis upon transduction with caCtnnb1, Ccnd2, carErbb2, or cahErbb2 LVs indicated induced CM proliferation.

### Lentiviral expression of mitogens in hiPSC-CMs may activate negative feedback loops

We then assessed molecular effects of LV transduction and found that caYap8SA-transduced hiPSC-CMs exhibited significantly increased gene expression of *Yap* and its downstream targets *Ctgf* and *Cyr61* (*Figure 1G*). However, the expression of the total and active, non-phosphorylated Yap

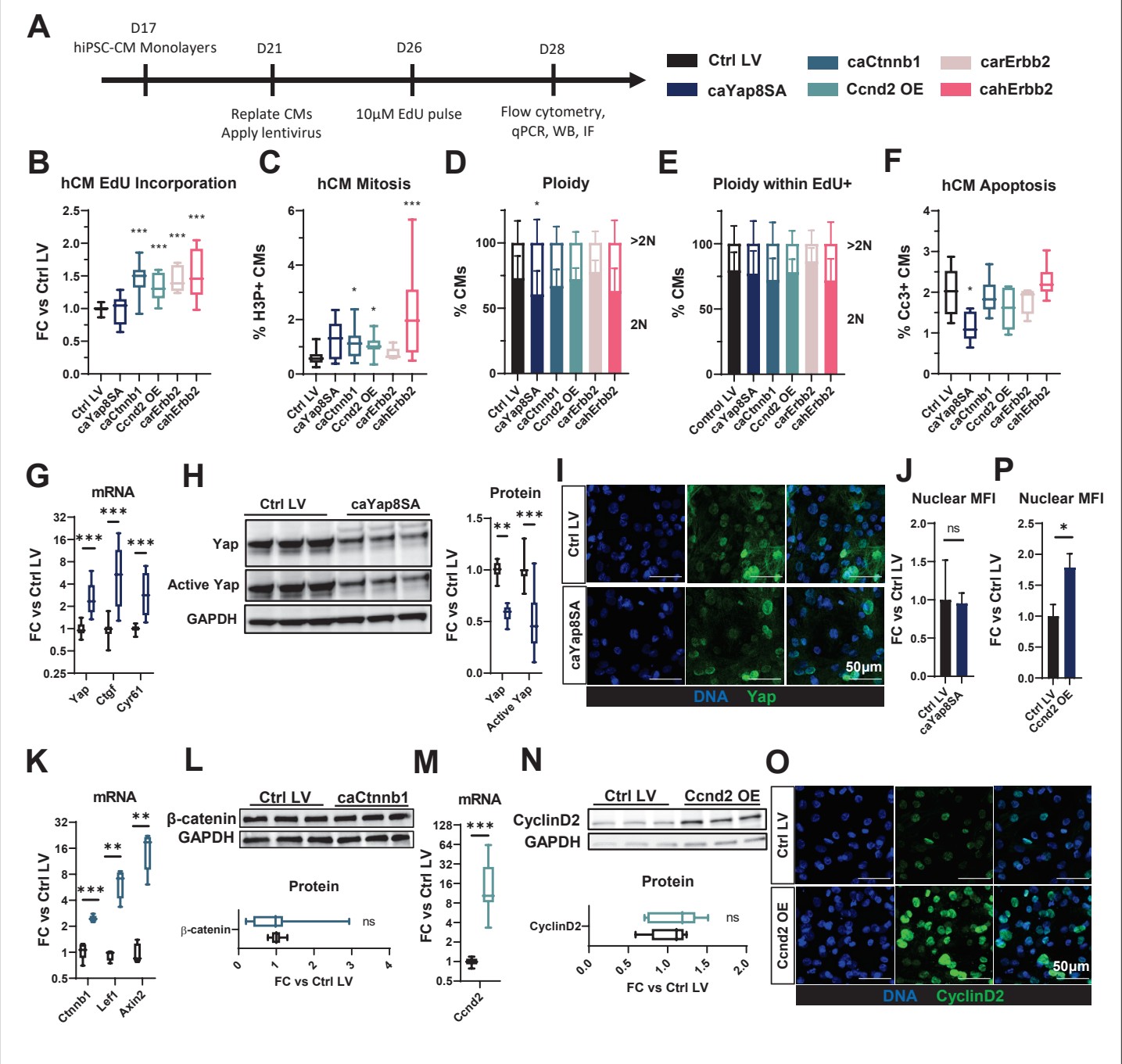

**Figure 1.** LV-delivered mitogens drive hiPSC-CM proliferation in monolayers. (**A**) Schematic of experimental design in hiPSC-CM monolayers. (**B–F**) Flow cytometry analysis of mCherry⁺ hiPSC-CMs showing (**B**) fold-change (FC) in EdU incorporation relative to control LV-transduced hiPSC-CMs, (**C**) percentage of H3P⁺ CMs, (**D, E**) percentage of 2 N and >2 N cells in all CMs (**D**) and EdU⁺ CMs (**E**), and (**F**) percentage of apoptotic Cc3⁺ CMs. (**G, H**) Analysis of relative (**G**) gene expression of *Yap* and its targets *Ctgf* and *Cyr61*, and (**H**) total and active Yap protein abundance in caYap8SA-transduced vs. control hiPSC-CMs. (**I,J**) Representative immunostaining images (**I**) and quantified nuclear mean fluorescence intensity (MFI; **J**) of YAP in caYap8SA-transduced versus control hiPSC-CMs. (**K, L**) Analysis of relative (**K**) expression of *Ctnnb1* and Wnt-signaling genes *Lef1* and *Axin2* and (**L**) Ctnnb1 protein abundance in caCtnnb1-transduced versus control hiPSC-CMs. (**M, N**) Analysis of relative Ccnd2 (**M**) gene and (**N**) protein expression and (**O, P**) representative immunostaining images (**O**) and quantified nuclear MFI (**P**) of Ccnd2 in Ccnd2-transduced versus control hiPSC-CMs. Data: box and whiskers showing distribution and min to max. Column graphs showing mean+ SD (*p<0.05, **p<0.01, ***p<0.001 vs. Ctrl LV). See *Supplementary file 1* for sample numbers and complete statistical information for all figures. Cc3, cleaved caspase-3; hiPSC-CM, human-induced pluripotent stem cell-derived cardiomyocyte; LV, lentiviral vector.

The online version of this article includes the following figure supplement(s) for figure 1:

*Figure 1 continued on next page*

*Figure 1 continued*

**Figure supplement 1.** Lentiviral design and transduction efficiency in CM monolayers.

**Figure supplement 2.** Flow cytometry method for analysis of CM proliferation and apoptosis.

**Figure supplement 3.** cahErbb2 induces cycling in low-glucose, palmitate-containing maturation culture media.

protein was significantly decreased (*Figure 1H*). Immunostaining of transduced hiPSC-CMs further revealed that the nuclear abundance of active Yap was unchanged by caYap8SA expression (*Figure 1I and J*). Because caYap8SA cannot be phosphorylated and degraded, these findings may be explained by the activation of a negative feedback loop that degraded endogenous Yap protein to compensate for caYap8SA overexpression, which in turn attenuated the mitogenic effects in caYap8SA-transduced hiPSC-CMs. We further assessed caCtnnb1-transduced hiPSC-CMs and found that the gene expression of *Ctnnb1* was increased, but also noted an increased expression of *Lef1* and *Axin2* (*Figure 1K*), potentially signifying the presence of a Wnt pathway-mediated negative feedback inhibitory to *Ctnnb1* expression (*Bernkopf et al., 2014*; *Lustig et al., 2002*). Supporting this hypothesis, we also observed no difference in Ctnnb1 protein expression between control and transduced cells (*Figure 1L*). Similar to caYap8SA and caCtnnb1, we further found increased Ccnd2 gene but not protein expression in Ccnd2-transduced hiPSC-CMs (*Figure 1M and N*). Still, immunostaining analysis revealed that Ccnd2-transduced hiPSC-CMs exhibited higher nuclear abundance of the protein (*Figure 1O and P*), which likely contributed to the observed increase in CM cycling (*Figure 1B and C*; *Zhu et al., 2018*). Overall, while these results suggest that LV-mediated expressions of various mitogens in hiPSC-CMs can induce post-transcriptional negative feedback responses to limit protein overexpression, mechanistic underpinnings of these processes remain to be additionally studied.

## cahErbb2 induces increased cell cycle activity in NRVM monolayers associated with sarcomere disassembly

To further compare the effects of studied mitogens in CMs from different species and maturity levels, we utilized cultured NRVMs. Specifically, NRVMs were transduced with the LVs the day of seeding and analyzed by flow cytometry at 14 days of monolayer culture, 2 days after EdU application (*Figure 2A*). We found that expression of cahErbb2 but not other mitogens resulted in increased EdU incorporation in NRVMs, while both Ccnd2 and cahErbb2 yielded increased H3P expression (*Figure 2B and C*). However, NRVMs transduced with some of the mitogens showed a significantly decreased fraction of diploid and increased fraction of polyploid CMs (*Figure 2D*), indicating that post-12 days of NRVM culture LV-induced cycling events led to CM polyploidization rather than cytokinesis, in contrast to the results in hiPSC-CMs (*Figure 1D*). This inference was supported by the finding that compared to control LV, the diploid fraction of EdU⁺ NRVMs was significantly decreased and polyploid fraction significantly increased in all LV-treated groups (*Figure 2E*). Taken together, these results revealed that by 14 days of 2D NRVM culture, cahErbb2 was still able to promote cell cycle entry and mitosis, while all tested mitogens led to increased NRVM polyploidy. Interestingly, in both NRVMs and hiPSC-CMs, LV expression of cahErbb2, but not other mitogens, also induced sarcomere disassembly (*Figure 2F*) without altering expression of sarcomeric genes (*Figure 2G*) or non-myocyte abundance (*Figure 2—figure supplement 1*). This result suggested that regulatory changes at a protein level were responsible for the loss of sarcomeric organization with cahErbb2 expression. Furthermore, increased expression of *Runx1* (*Figure 2G*) implied that cell cycle activity and sarcomere loss in cahErbb2-expressing NRVMs were associated with cell dedifferentiation (*D'Uva et al., 2015*; *Kubin et al., 2011*).

## cahErbb2 induces cell cycle activation, growth, and contractile deficit in NRVM cardiobundles

We further tested whether LV delivery of mitogens affected CM proliferation and function in three-dimensional NRVM cardiobundles, currently representing the most mature in vitro model of the post-natal myocardium (; *Jackman et al., 2016*). NRVMs were transduced at the time of cardiobundle formation, pulsed with EdU at culture day 12, and assessed after 2 weeks of culture (*Figure 3A*), first ensuring that no studied structural or functional properties differed between control LV-transduced and non-transduced tissues (*Figure 3—figure supplement 1*). From cardiobundle cross-sectional stainings, EdU⁺ nuclei were observed in both vimentin⁺ cardiac fibroblasts (; *Jackman et al., 2016*)

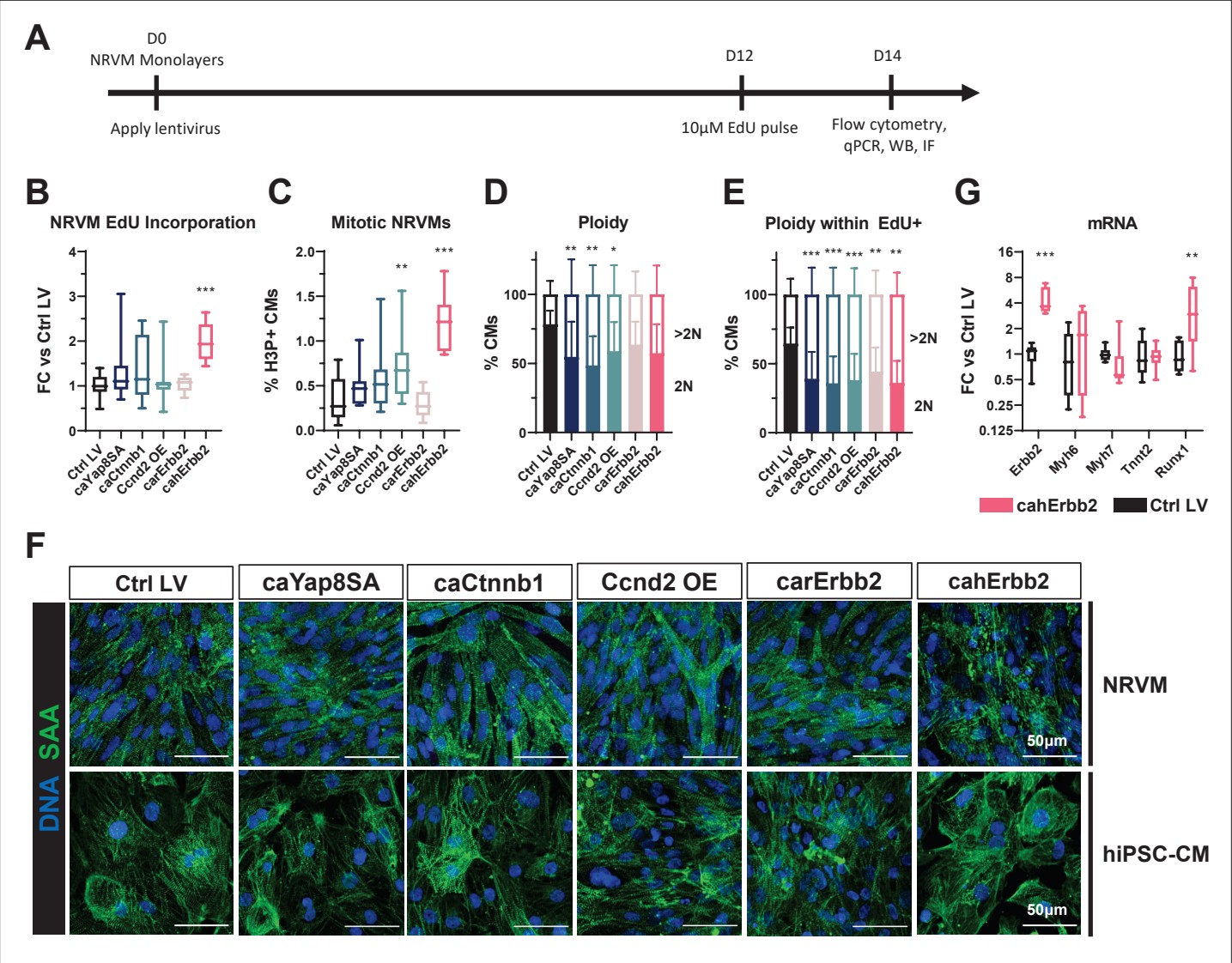

**Figure 2.** cahErbb2 induces NRVM cycle entry in monolayers and promotes sarcomere disassembly in NRVMs and hiPSC-CMs. (**A**) Schematic of experimental design in NRVM monolayers. (**B–E**) Flow cytometry analysis of mCherry⁺ NRVMs showing (**B**) fold-change (FC) in EdU incorporation relative to control LV-treated NRVMs, (**C**) percentage of H3P⁺ CMs, and (**D, E**) percentage of 2 N and >2 N cells in all CMs (**D**) and EdU⁺ CMs (**E**). (**F**) Representative immunostaining images of sarcomeric α-actinin showing sarcomeric structure in LV transduced NRVMs and hiPSC-CMs. (**G**) Relative expression of *Erbb2*, sarcomeric genes (*Myh6*, *Myh7*, and *Tnnt2*), and dedifferentiation marker *Runx1* in cahErbb2-transduced versus control hiPSC-CMs. Data: box and whiskers showing distribution and min to max. Column graphs showing mean+ SD (*p<0.05, **p<0.01, ***p<0.001 vs. Ctrl LV). hiPSC-CM, human-induced pluripotent stem cell-derived cardiomyocyte; LV, lentiviral vector; NRVM, neonatal rat ventricular myocyte.

The online version of this article includes the following figure supplement(s) for figure 2:

**Figure supplement 1.** cahErbb2 expression does not affect non-CM abundance or CM size in hiPSC-CM and NRVM monolayers.

predominantly residing at the tissue periphery and in F-actin⁺/vimentin⁻ CMs (*Figure 3B*, top, middle). Similar to hiPSC-CM and NRVM monolayers, all studied mitogens except caYap8SA increased EdU incorporation in NRVM cardiobundles, with cahErbb2 showing the strongest effect (*Figure 3C*). We then assessed morphological and functional characteristics of cardiobundles and found that cahErbb2 expression uniquely increased both the total cross-sectional area (CSA; *Figure 3D*) and F-actin⁺ (CM) area of cardiobundles, leading to the formation of a necrotic core (devoid of Hoechst-positive nuclei; *Figure 3B and E*), likely caused by limited diffusion of oxygen and nutrients into the center of these avascular tissues (*Laschke and Menger, 2016*). We also observed increased vimentin⁺ CSA, indicating increased fibroblast abundance (*Figure 3F*) and by dividing F-actin⁺/vimentin⁻ CSA with number

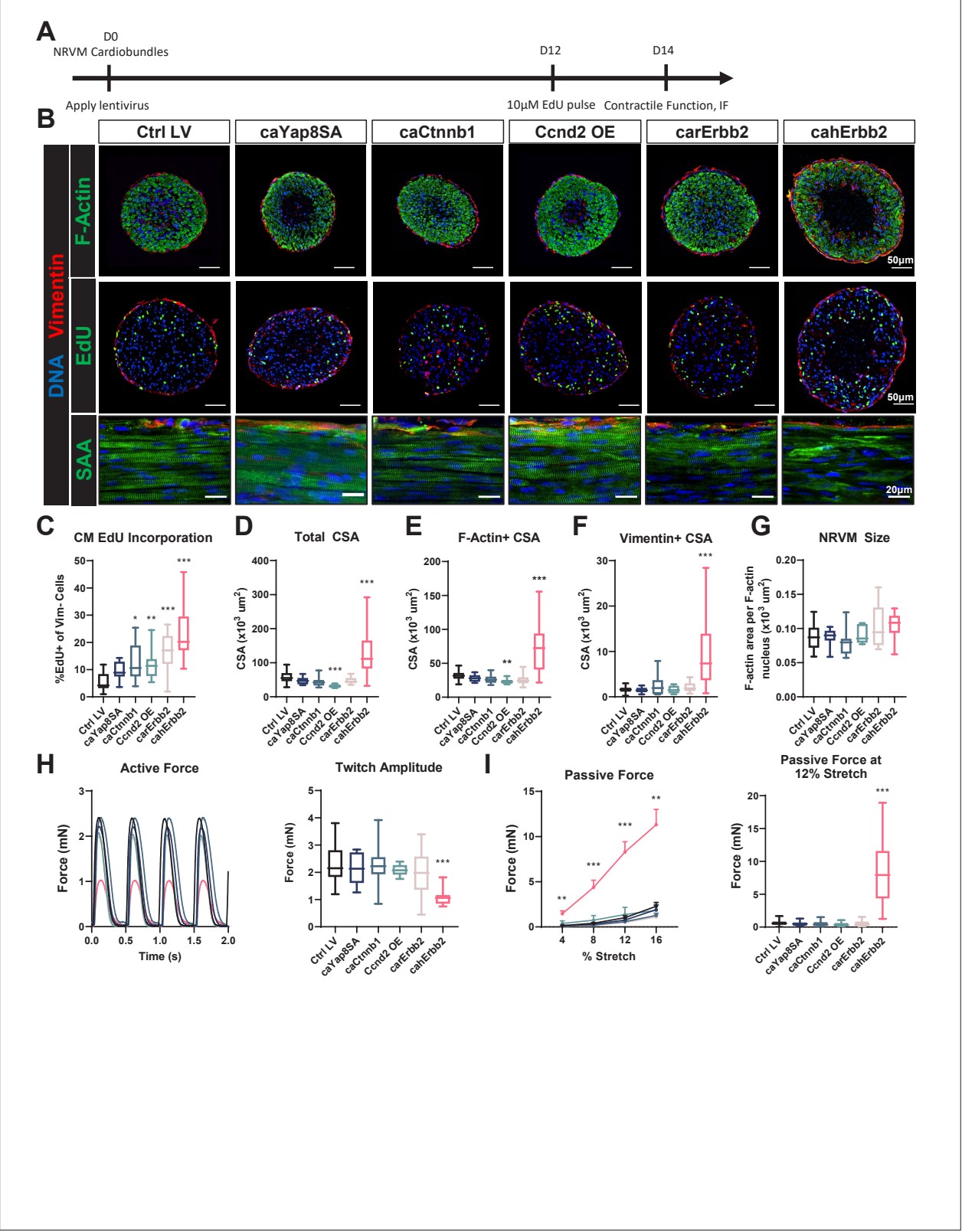

**Figure 3.** cahErbb2 induces NRVM cycle entry in cardiobundles and promotes sarcomere disassembly and contractile dysfunction. (**A**) Schematic of experimental design in NRVM cardiobundles. (**B**) Representative immunostaining images of cardiobundle cross-sections showing morphology (top), EdU incorporation (middle), and whole-mount tissue showing sarcomere structure (bottom). (**C–G**) Quantification of immunostained cardiobundle cross-sections for (**C**) NRVM EdU incorporation and (**D**) total, (**E**) F- actin+, (**F**) vimentin+ cross-sectional area (CSA), and (**G**) Quantified F-actin+ CSA per nuclei

*Figure 3 continued on next page*

*Figure 3 continued*

number within this area, shown as a measure of CM size. (**H, I**) Force analysis in LV-transduced cardiobundles showing (**H**) representative twitch traces and quantified maximum twitch amplitude and (**I**) passive force-length and force amplitude at 12% stretch (0% stretch is culture length of 7 mm). Data: box and whiskers showing distribution and min to max. Line plot showing mean+ SEM (*p<0.05, **p<0.01, ***p<0.001 vs. Ctrl LV). LV, lentiviral vector; NRVM, neonatal rat ventricular myocyte.

The online version of this article includes the following figure supplement(s) for figure 3:

**Figure supplement 1.** LV transduction does not affect NRVM cardiobundles.

**Figure supplement 2.** Twitch kinetics and unloaded contractile force in cardiobundles.

**Figure supplement 3.** Underlying mechanisms of increased passive tension in cahErbb2-expressing cardiobundles.

of nuclei within that area inferred that CM size was not significantly altered by mitogen expression (*Figure 3G*). In CM monolayers, we also compared relative cell size using flow cytometry and found that cahErbb2 does not affect CM size (*Figure 2—figure supplement 1D, E*). Coupled with our measurements in cardiobundles, this confirmed that cahErbb2 expression does not increase CM size in either culture setting or cell type. Finally, similar to the findings in CM monolayers (*Figure 2G*), only the cahErbb2-transduced cardiobundles showed a near-complete loss of sarcomere structure, despite maintained NRVM alignment (*Figure 3B*, bottom).

We then measured contractile force generation in cardiobundles and found a reduced maximum twitch amplitude in tissues transduced with cahErbb2 but not other mitogens (*Figure 3H*). This loss of maximum active force in cahErbb2 cardiobundles was accompanied by slower twitch kinetics (*Figure 3—figure supplement 2A-C*), as well as a significant increase in passive tension that indicated increased tissue stiffness (*Figure 3I*). We then confirmed that reduced contractile force generation due to cahErbb2 expression was also characteristic of mechanically unloaded tissues (*Figure 3—figure supplement 2D, E*). To examine the potential causes of cahErbb2-induced increase in stiffness, we first generated tissues with 1.4× NRVM/hydrogel volume which exhibited an increased size and necrotic core similar to those of cahErbb2 cardiobundles, however, this did not lead to larger passive tension (*Figure 3—figure supplement 3A*). We compared the 1.4× tissues to cahErbb2 tissues and found that both the tissue size and acellular area within the tissues were similar, which suggests that a larger tissue alone does not cause greatly increased passive tension as seen with cahErbb2 cardiobundles (*Figure 3—figure supplement 3B*). We further examined collagen I in cross-sections of cahErbb2 cardiobundles and found similar abundance to that of control tissues, indicating that increased tissue stiffness was not contributed by significant ECM accumulation (*Figure 3—figure supplement 3C*; *Li et al., 2017*). However, we found significant upregulation of intermediate filaments nestin and vimentin (*Figure 3F*, *Figure 3—figure supplement 3D, E*), suggesting that cahErbb2-induced cell stiffening, at least partly, was responsible for the observed increase in tissue stiffness (*Calderone, 2018*; *Ward and Iskratsch, 2020*). Taken together, among the studied mitogens, cahErbb2 induced the most potent pro-proliferative effects in both rat and human CMs, which also involved the sarcomere loss characteristic of CM dedifferentiation observed in carErbb2-expressing mice (*D'Uva et al., 2015*). Furthermore, in engineered NRVM cardiobundles, cahErbb2 mitogenic effects were uniquely associated with the increase in tissue size, loss of contractile force, increased intermediate filament expression, and tissue stiffening.

## Human but not rat caErbb2 activates Erk signaling in hiPSC-CMs

We were intrigued by the finding that human but not rat caErbb2 exerted significant mitogenic effects in hiPSC-CMs and NRVMs, and decided to further probe the mechanisms of cahErbb2 action. Previously, the expression of rat caErbb2 in mouse CMs led to sarcomere disassembly in vitro and in vivo, while downstream Erk and Akt signaling were found to be the primary drivers of carErbb2-induced CM cell cycle activation (*D'Uva et al., 2015*). In hiPSC-CMs in our study, cahErbb2 but not carErbb2 expression resulted in increased pErk and Erk abundance (*Figure 4A*). Moreover, previously reported pAkt increase was not observed for either rat or human caErbb2 expression, with carErbb2 expression reducing total Akt (*Figure 4A*). We then measured the relative gene expression of Erk downstream targets indicative of increased Erk activity (*Wagle et al., 2018*). Whereas cahErbb2 expression robustly increased transcription of multiple Erk targets, carErbb2 expression caused no such increase (*Figure 4B*). Consistent with the cahErbb2-induced Erk activation, we also observed an increase in

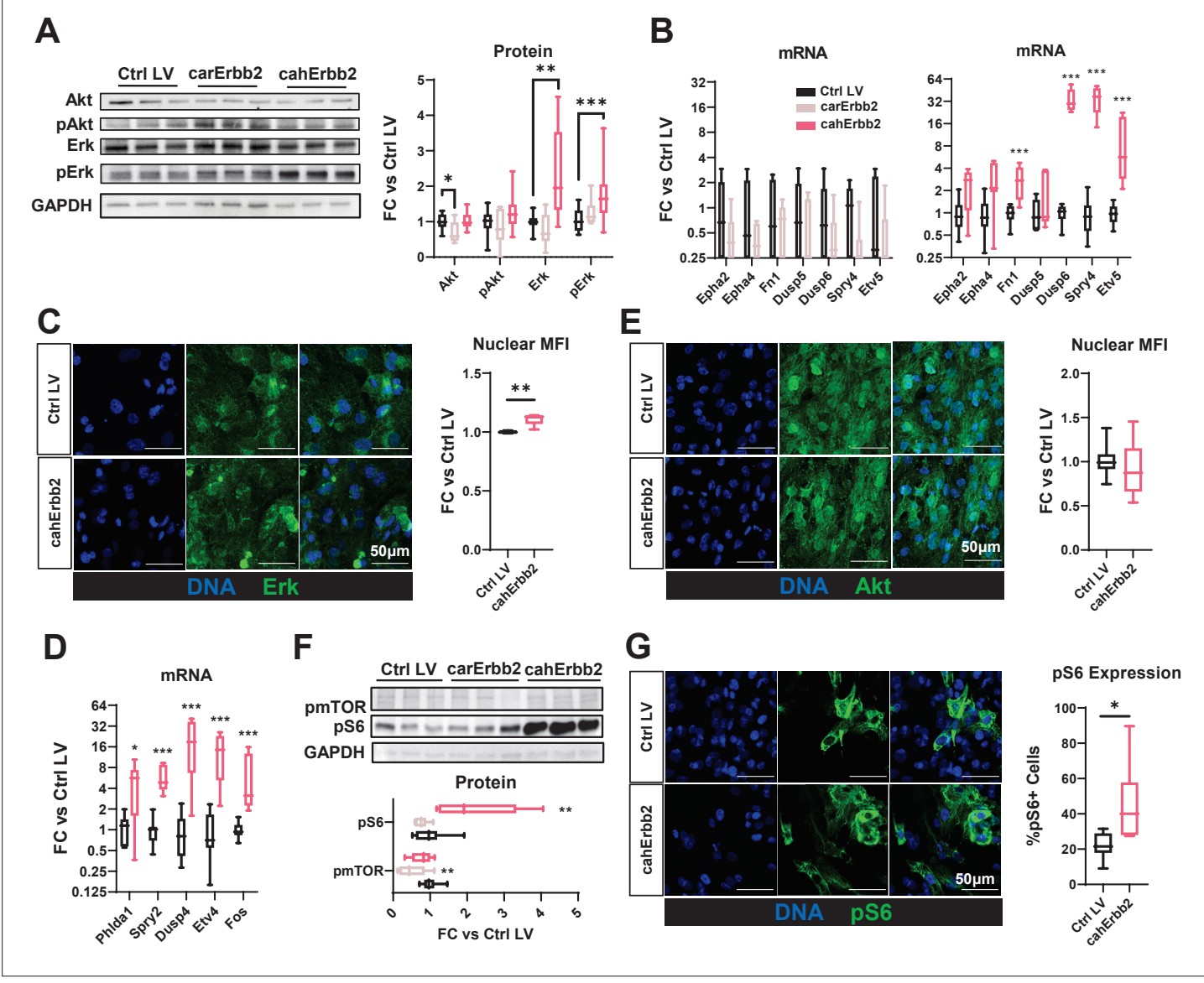

**Figure 4.** cahErbb2 but not carErbb2 activates Erk signaling to drive proliferation in CMs. (**A–B**) Representative Western blots and quantified relative protein (**A**) and Erk target gene expression (**B**) in carErbb2- or cahErbb2-transduced versus control hiPSC-CM monolayers. (**C**) Representative immunostaining images and quantified nuclear MFI of Erk in cahErbb2-transduced versus control hiPSC-CMs. (**D**) Quantified relative Erk target gene expression in cahErbb2-transduced versus control hiPSC-CM monolayers. (**E**) Representative immunostaining images and quantified nuclear MFI of Akt in cahErbb2-transduced versus control hiPSC-CMs. (**F**) Representative Western blots and quantified relative phosphorylated mTOR (pmTOR) and ribosomal protein S6 (pS6) expression in carErbb2- or cahErbb2-transduced versus control hiPSC-CMs. (**G**) Representative immunostaining images and quantified nuclear MFI of pS6 in cahErbb2-transduced versus control hiPSC-CMs. Data: box and whiskers showing distribution and min to max. Column graphs showing mean+ SD (*p<0.05, **p<0.01, ***p<0.001 vs. Ctrl LV). hiPSC-CM, human-induced pluripotent stem cell-derived cardiomyocyte; LV, lentiviral vector MFI, mean fluorescence intensity.

nuclear localization of Erk protein (*Figure 4C*), and further confirmed Erk activation by analyzing the expression of additional gene targets (*Figure 4D*). In contrast to Erk, Akt localization was unaffected by cahErbb2 expression (*Figure 4E*). Interestingly, cahErbb2 but not carErbb2 also increased the abundance of phosphorylated ribosomal protein S6 (pS6) in Western blots (*Figure 4F*), which was further confirmed by immunostaining (*Figure 4G*). pS6 is usually considered as a downstream target of mTOR, which can be activated by stimulation of Akt or Erk pathway (*Roux et al., 2007*; *Warfel et al., 2011*). Curiously, pmTOR expression was unaffected by cahErbb2 (*Figure 4F*) and the Akt pathway was not upregulated (*Figure 4A and B*); thus, the human caErbb2-induced pS6 increase

likely resulted from an mTOR-independent, Erk-dependent mechanism previously associated with Ser235/236 phosphorylation of pS6 (*Roux et al., 2007*).

## Erk or Mek inhibition attenuates cahErbb2-induced effects in hiPSC-CMs and NRVMs

To determine whether upregulated Erk signaling was required for the observed effects of cahErbb2 expression on hiPSC-CMs, we performed flow cytometry analysis in cells treated with the Mek inhibitor (Meki) PD0325901 (Mirdametinib) or the Erk inhibitor (Erki) SCH772984 applied for 48 hr before sample collection; EdU was applied during the final 24 hr to capture DNA synthesis that only occurred after adding the inhibitors (*Hashimoto et al., 2014*). We simultaneously measured Cc3 abundance because excessive Erk inhibition was expected to interfere with homeostatic Erk activity required to promote cell survival (*Lu and Xu, 2006*). As expected, in both control and cahErbb2-treated hiPSC-CMs, Meki or Erki treatment resulted in a dose-dependent decrease in EdU incorporation and a dose-dependent increase or an increasing trend in apoptotic events (*Figure 5—figure supplement 1*). The increased Erk activity in cahErbb2-transduced CMs appeared to both protect against the Erki/Meki-induced apoptosis and necessitate higher inhibitor doses to maximally block EdU incorporation (*Figure 5—figure supplement 1*). We then tested whether the inhibition of Erk signaling pathway can prevent cahErbb2-induced effects in NRVM cardiobundles by applying 100 nM Erki or 100 nM Meki between days 8 and 14 of culture (*Figure 5A*). We found that both inhibitors reduced CM EdU incorporation in control and cahErbb2-transduced cardiobundles, while the Erki-treated cahErbb2 tissues still showed higher rates of cycling compared to the control (*Figure 5B*, middle, *Figure 5C*). Strikingly, the Erk and Mek inhibition also attenuated cahErbb2-induced deficits in tissue morphology, evident from the findings that inhibitor-treated, cahErbb2 cardiobundles had total, F-actin$^+$, and vimentin$^+$ CSA comparable to those of control cardiobundles (*Figure 5B and D–F*) and showed improved sarcomere structure, especially with the Erki treatment (*Figure 5B*). Furthermore, while the inhibitors did not affect force responses in the control cardiobundles, inhibitor-treated cahErbb2 tissues exhibited higher active force (*Figure 5G*) and lower passive tension (*Figure 5H*), as well as improved twitch kinetics (*Figure 5—figure supplement 2*) compared to vehicle treatment. Collectively, these results suggested that the baseline CM cycling in NRVM cardiobundles is Erk-dependent and that cahErbb2-induced morphological and functional deficits can be prevented or reverted by Erk/Mek inhibition, further providing evidence that Erk pathway is a major effector of cahErbb2 expression in CMs.

## Discussion

Development of methods to induce transient proliferation of endogenous or transplanted CMs could lead to therapeutic strategies to restore lost muscle mass and function of the injured heart (*Karra and Poss, 2017*). In this in vitro study, we directly compared a panel of lentivirally delivered mitogens to identify those capable of inducing the most potent and conserved pro-proliferative effects in CMs from different species, maturation stages, and culture environments. We found that constitutively active human Erbb2 induced the most robust and consistent proliferative response in hiPSC-CMs and NRVMs among rat caErbb2, human caCtnnb1, human Ccnd2, and human caYap8SA. Interestingly, we found evidence for the existence of negative post-transcriptional feedbacks on expression of mitogen proteins following transduction with caYap8SA, caCtnnb1, or Ccnd2 LVs, which appeared strongest in caYap8SA-transduced rat and human CMs where we did not find increased EdU incorporation. A negative feedback mechanism in the Hippo pathway mediated by Lats1/2 upregulation was previously reported when constitutively active Yap was expressed or Sav1 was deleted in mice and cultured cells (*Kim et al., 2020*; *Moroishi et al., 2015*; *Park et al., 2016*). Regardless, our findings appear to contradict the potent proliferative effects of constitutively active Yap mutants observed in previous studies (*Mills et al., 2017*; *Monroe et al., 2019*; *von Gise et al., 2012*), which may be attributable to differences in species and mechanical environment of studied CMs (*Aragona et al., 2013*; *Benham-Pyle et al., 2015*; *Meng et al., 2018*; *Panciera et al., 2020*; *Zhao et al., 2007*), and will require further investigations.

Although cahErbb2 caused strong mitogenic effects in CMs, we suspect that this did not translate to improved contractile function in our cardiobundle system for two reasons. First, cell survival in the avascular cardiobundles relies on the ability of oxygen and nutrients to diffuse from the surface to

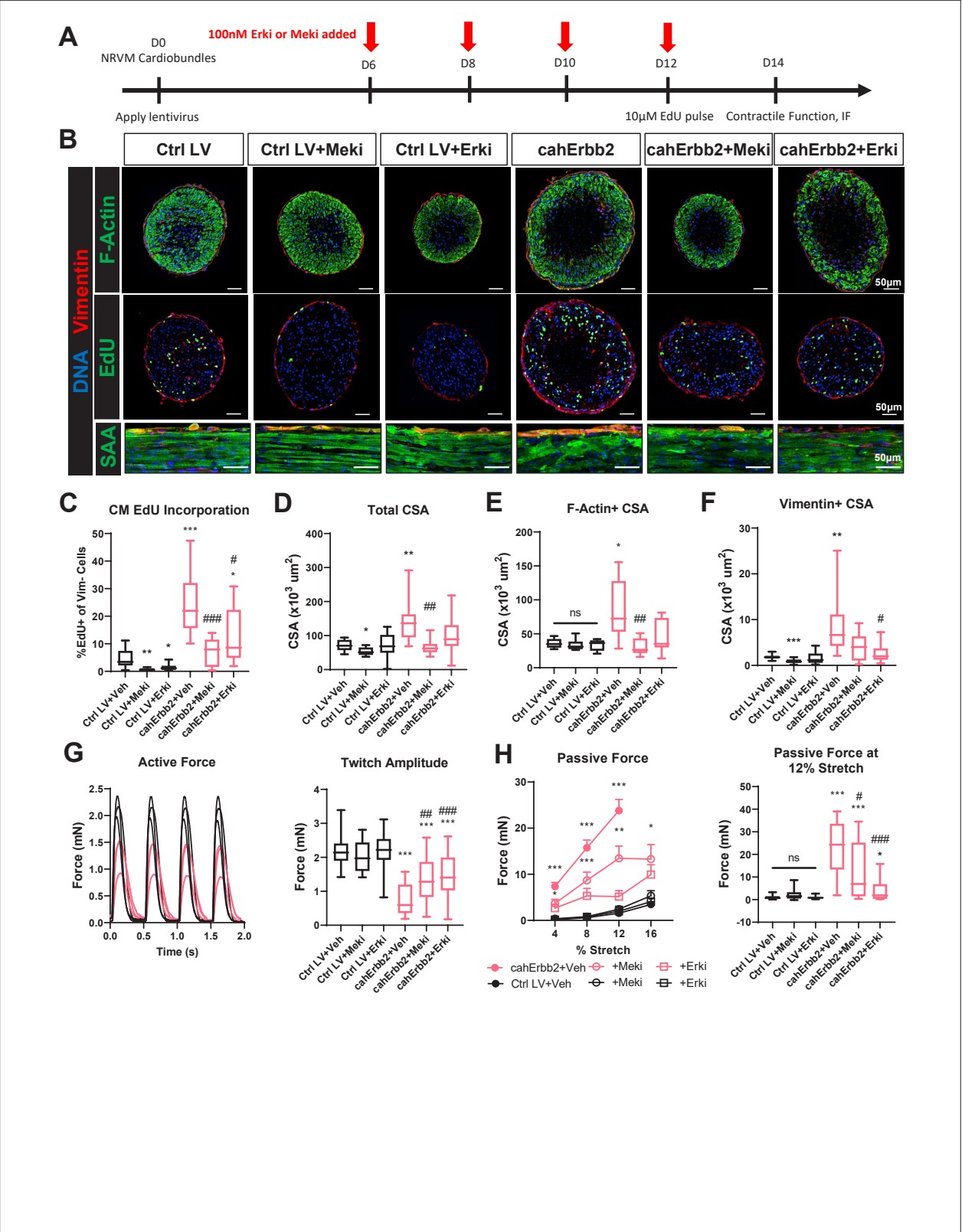

**Figure 5.** Erk or Mek inhibition attenuates cahErbb2-induced effects in NRVM cardiobundles. (**A**) Schematic of experimental design in cahErbb2 and Ctrl LV NRVM cardiobundles. (**B**) Representative immunostaining images of cardiobundle cross-sections showing morphology (top), EdU incorporation (middle), and whole-mount tissue showing sarcomere structure (bottom). (**C–F**) Quantification of immunostained cardiobundle cross-sections for (**C**) NRVM EdU incorporation and (**D**) total, (**E**) F- actin⁺, and (**F**) vimentin⁺ cross-sectional area (CSA). (**G, H**) Force analysis in cardiobundles showing (**G**)

*Figure 5 continued on next page*

*Figure 5 continued*

representative twitch traces and quantified maximum twitch amplitude and (**H**) passive force-length and force amplitude at 12% stretch ( 0% stretch is culture length of 7 mm). Data: box and whiskers showing distribution and min to max. Line plot showing mean+ SEM (\*p<0.05, \*\*p<0.01, \*\*\*p<0.001 vs. Ctrl LV; #p<0.05, ##p<0.01, ###p<0.001 vs. cahErbb2). LV, lentiviral vector; NRVM, neonatal rat ventricular myocyte.

The online version of this article includes the following figure supplement(s) for figure 5:

**Figure supplement 1.** Erk or Mek inhibition attenuates cahErbb2-induced proliferation in hiPSC-CMs.

**Figure supplement 2.** Twitch kinetics of NRVM cardiobundles treated with Erki or Meki.

the center of the tissue (***Bursac et al., 1999***; ***Laschke and Menger, 2016***). This limits the growth of metabolically active, functional myocardium and causes the development of central necrotic core, as observed in both cahErbb2-transduced and control, larger-volume tissues. Second, the LV-induced, continuous cahErbb2 expression in cardiobundles not only produces persistent proliferative effects, but also chronically over-activates Erk pathway leading to sarcomere disorganization and increased tissue stiffness. The partial functional rescue with Erki/Meki treatment suggests that in the vascularized myocardium in vivo, temporally controlled rather than the continuous expression of cahErbb2 could be a strategy to induce both transient CM hyperplasia and, ultimately, following cahErbb2 inactivation, enhanced myocardial function.

We were surprised to find that only the human but not rat ortholog of caErbb2 was able to activate Erk signaling in hCMs and NRVMs, especially considering the rat Erbb2-V663E and human Erbb2-V659E activating mutations are both located in the transmembrane domain and known to promote homo- and hetero-dimerizations of the receptor (***Ou et al., 2017***). Previously, rat caErbb2 has been shown to drive CM dedifferentiation, proliferation, and hypertrophy when inducibly expressed in transgenic mice (***Aharonov et al., 2020***; ***D'Uva et al., 2015***). In our study, persistent cahErbb2 expression and downstream Erk activation induced not only CM cycling but also structural (sarcomere disassembly and fibroblast accumulation) and functional (reduced contractile force and increased stiffness) deficits, which in several aspects resembled in vivo pathologic cardiac remodeling mediated by hyperactive Erk signaling (***Gallo et al., 2019***). While we noticed increased *Runx1* expression in cahErbb2-transduced hiPSC-CMs, we did not observe a gene expression pattern characteristic of sustained CM de-differentiation as *Myh6, Myh7*, and *Tnnt2* levels remained unchanged. Although our in vitro systems lack the complexity of an in vivo cardiac milieu (e.g., cellular diversity and related signaling cross-talk), it is possible that the effects of caErbb2 expression in CMs may be species or maturation level-dependent. Further work will be needed to explore if human caErbb2 or its downstream effectors can generate comparable or more potent therapeutic effects than those of rat caErbb2 in both small and large animal studies.

## Materials availability

Plasmids generated in this study are available from the corresponding author upon reasonable request, maps of generated plasmids are provided in ***Source data 4***.

## Materials and methods

**Key resources table**

| Reagent type (species) or resource | Designation | Source or reference | Identifiers | Additional information |
|---|---|---|---|---|
| Antibody | Anti-Cardiac Troponin T (Rabbit polyclonal) | Abcam | ab45932 | FC (1:200) |
| Antibody | Anti-Sarcomeric Alpha Actinin (Mouse monoclonal) | Sigma-Aldrich | A7811 | IF (1:200) |
| Antibody | Cleaved Caspase-3 (Asp175) (Rabbit polyclonal) | Cell Signaling Technology | 9661 | FC (1:800) |
| Antibody | GAPDH (Mouse monoclonal) | Santa Cruz Biotechnology | sc-47724 | WB (1:1000) |

*Continued on next page*

*Continued*

| Reagent type (species) or resource | Designation | Source or reference | Identifiers | Additional information |
|---|---|---|---|---|
| Antibody | Mouse IgG1, kappa monoclonal [15-6E10A7] - Isotype Control | Abcam | ab170190 | FC (1:2000) |
| Antibody | Phospho-p44/42 MAPK (Erk1/2) (Thr202/Tyr204) (Rabbit polyclonal) | Cell Signaling Technology | 9101 | WB (1:1000) |
| Antibody | p44/42 MAPK (Erk1/2) (Total Erk) (Rabbit polyclonal) | Cell Signaling Technology | 9102 | WB (1:1000) |
| Antibody | Phospho-Histone H3 (Ser10) (6G3) (Mouse monoclonal) | Cell Signaling Technology | 9706 | FC (1:1000) |
| Antibody | Phospho-mTOR (Ser2448) (Rabbit polyclonal) | Cell Signaling Technology | 2971 | WB (1:1000) |
| Antibody | Phospho-S6 Ribosomal Protein (Ser235/236) (Rabbit polyclonal) | Cell Signaling Technology | 2211 | WB (1:1000) IF (1:100) |
| Antibody | Rabbit IgG, polyclonal - Isotype Control | Abcam | ab37415 | FC (1:2000) |
| Antibody | Recombinant Anti-active YAP1 [EPR19812] (Rabbit polyclonal) | Abcam | ab205270 | WB (1:1000) IF (1:500) |
| Antibody | Recombinant Anti-AKT1 (phospho S473) [EP2109Y] (Rabbit monoclonal) | Abcam | ab81283 | WB (1:5000) |
| Antibody | Recombinant Anti-AKT1+ AKT2+ AKT3 [EPR16798] (Rabbit monoclonal) | Abcam | ab179463 | WB (1:10,000) IF (1:100) |
| Antibody | Recombinant Anti-Cyclin D2 (Rabbit monoclonal) | Abcam | ab207604 | WB (1:1000) IF (1:100) |
| Antibody | Recombinant Anti-Vimentin [EPR3776] (Rabbit monoclonal) | Abcam | ab92547 | IF (1:500) |
| Antibody | YAP (D8H1X) (Total Yap) (Rabbit monoclonal) | Cell Signaling Technology | 14074 | WB (1:1000) IF (1:200) |
| Antibody | β-Catenin (Carboxy-terminal Antigen) (Rabbit polyclonal) | Cell Signaling Technology | 9587 | WB (1:1000) |
| Chemical compound, drug | SCH772984 | Cayman Chemical | 19166 | |
| Chemical compound, drug | PD0325901 | Sigma-Aldrich | PZ0162-5MG | |
| Commercial assay, kit | Click-iT EdU Alexa Fluor 647 Flow Cytometry Assay Kit | Thermo Fisher Scientific | C10419 | |
| Commercial assay, kit | Click-iT EdU Alexa Fluor 488 Imaging Kit | Thermo Fisher Scientific | C10337 | |
| Cell line (*Homo sapiens*, male) | DU11 iPSC Line | Duke University Stem Cell Core | | |
| Cell line (*H. sapiens*) | Hek293T | ATCC | CRL-3216 | |
| Strain, strain background (*Rattus Norvegicus*) | P2 Sprague-Dawley Rat Pups | Charles River | | |

*Continued on next page*

*Continued*

| Reagent type (species) or resource | Designation | Source or reference | Identifiers | Additional information |
|---|---|---|---|---|
| Recombinant DNA reagent | pLV-beta-catenin deltaN90 | pLV-beta-catenin deltaN90 was a gift from Bob Weinberg | Addgene plasmid # 36985 | Used to generate new plasmids |
| Recombinant DNA reagent | HER2 CA (V659E) | HER2 CA (V659E) was a gift from Mien-Chie Hung | Addgene plasmid # 16259 | Used to generate new plasmids |
| Recombinant DNA reagent | R777-E020 Hs.CCND2-nostop | R777-E020 Hs.CCND2-nostop was a gift from Dominic Esposito | Addgene plasmid # 70304 | Used to generate new plasmids |
| Recombinant DNA reagent | pSV2 neuNT | pSV2 neuNT was a gift from Bob Weinberg | Addgene plasmid # 10,918 | Used to generate new plasmids |
| Recombinant DNA reagent | pCMV-flag YAP2 5SA | pCMV-flag YAP2 5SA was a gift from Kunliang Guan | Addgene plasmid # 27371 | Used to generate new plasmids |
| Recombinant DNA reagent | Pax2 | psPAX2 was a gift from Didier Trono | Addgene plasmid # 12260 | Used to generate lentivirus in HEK293Ts |
| Recombinant DNA reagent | VSVG | pMD2.G was a gift from Didier Trono | Addgene plasmid # 12259 | Used to generate lentivirus in HEK293Ts |
| Recombinant DNA reagent | Control Lentivirus (MHCK7-mCherry) | This manuscript | | See Materials and methods section |
| Recombinant DNA reagent | Cyclin Lentivirus (MHCK7-Ccnd2-P2A-mCherry-NLS) | This manuscript | | See Materials and methods section |
| Recombinant DNA reagent | B-catenin Lentivirus (MHCK7-Ctnnb1(Δ90)-P2A-mCherry-NLS) | This manuscript | | See Materials and methods section |
| Recombinant DNA reagent | Human Erbb2 Lentivirus (MHCK7-Erbb2(V659E-P2A-mCherry)) | This manuscript | | See Materials and methods section |
| Recombinant DNA reagent | Rat Erbb2 Lentivirus (MHCK7-Erbb2(V663E-P2A-mCherry-NLS)) | This manuscript | | See Materials and methods section |
| Recombinant DNA reagent | Yap Lentivirus (MHCK7-Yap8SA-P2A-mCherry-NLS) | This manuscript | | See Materials and methods section |
| Software, algorithm | CellProfiler | *McQuin et al., 2018* | https://cellprofiler.org/ | |
| Software, algorithm | ImageJ | *Schneider et al., 2012* | https://imagej.nih.gov/ij/ | |
| Software, algorithm | FlowJo Software | *The FlowJo team, 2021* | | |
| Software, algorithm | GraphPad Prism | | https://www.graphpad.com/ | |
| Software, algorithm | Matlab | | | |
| Sequence-based reagent | Yap1 | | PrimerBank ID: 303523510 c1 | F:TAGCCCTGCGTAGCCAGTTA R:TCATGCTTAGTCCACTGTCTGT |
| Sequence-based reagent | Myh6 | | NCBI PrimerBlast | F:GCCCTTTGACATTCGCACTG R:GGTTTCAGCAATGACCTTGCC |
| Sequence-based reagent | Myh7 | | PrimerBank ID: 115496168 c1 | F:ACTGCCGAGACCGAGTATG R:GCGATCCTTGAGGTTGTAGAGC |
| Sequence-based reagent | Ctgf | | PrimerBank ID: 98986335 c1 | F:CAGCATGGACGTTCGTCTG R:AACCACGGTTTGGTCCTTGG |
| Sequence-based reagent | Cyr61 | | PrimerBank ID: 197313774 c3 | F:CAGCATGGACGTTCGTCTG R:AACCACGGTTTGGTCCTTGG |
| Sequence-based reagent | Ctnnb1 | | PrimerBank ID: 148233337 c2 | F:CATCTACACAGTTTGATGCTGCT R:GCAGTTTTGTCAGTTCAGGGA |

*Continued on next page*

*Continued*

| Reagent type (species) or resource | Designation | Source or reference | Identifiers | Additional information |
|---|---|---|---|---|
| Sequence-based reagent | Lef1 | | PrimerBank ID: 260656055 c1 | F:AGAACACCCCGATGACGGA R:GGCATCATTATGTACCCGGAAT |
| Sequence-based reagent | Axin2 | | PrimerBank ID: 195927058 c1 | F:CAACACCAGGCGGAACGAA R:GCCCAATAAGGAGTGTAAGGACT |
| Sequence-based reagent | Ccnd2 | | PrimerBank ID: 209969683 c3 | F:TTTGCCATGTACCCACCGTC R:AGGGCATCACAAGTGAGCG |
| Sequence-based reagent | Erbb2 | | PrimerBank ID: 54792097 c2 | F:TGTGACTGCCTGTCCCTACAA R:CCAGACCATAGCACACTCGG |
| Sequence-based reagent | Epha2 | | PrimerBank ID: 296010835 c2 | F:AGAGGCTGAGCGTATCTTCAT R:GGTCCGACTCGGCATAGTAGA |
| Sequence-based reagent | Epha4 | | PrimerBank ID: 45439363 c3 | F:GCAAGGAGACGTTTAACCTGT R:CTTGGGTGAAGCTCTCATCAG |
| Sequence-based reagent | Fn1 | | PrimerBank ID: 47132556 c2 | F:AGGAAGCCGAGGTTTTAACTG R:AGGACGCTCATAAGTGTCACC |
| Sequence-based reagent | Dusp5 | | PrimerBank ID: 62865889 c2 | F:GCCAGCTTATGACCAGGGTG R:GTCCGTCGGGAGACATTCAG |
| Sequence-based reagent | Dusp6 | | PrimerBank ID: 42764682 c1 | F:GAAATGGCGATCAGCAAGACG R:CGACGACTCGTATAGCTCCTG |
| Sequence-based reagent | Spry4 | | PrimerBank ID: 188595696 c1 | F:TCTGACCAACGGCTCTTAGAC R:GTGCCATAGTTGACCAGAGTC |
| Sequence-based reagent | Etv5 | | PrimerBank ID: 194018465 c1 | F:TCAGCAAGTCCCTTTTATGGTC R:GCTCTTCAGAATCGTGAGCCA |
| Sequence-based reagent | Phlda1 | | PrimerBank ID: 83977458 c1 | F:GAAGATGGCCCATTCAAAAGCG R:GAGGAGGCTAACACGCAGG |
| Sequence-based reagent | Spry2 | | PrimerBank ID: 22209007 c1 | F:CCTACTGTCGTCCCAAGACCT R:GGGGCTCGTGCAGAAGAAT |
| Sequence-based reagent | Dusp4 | | PrimerBank ID: 325651887 c1 | F:GGCGGCTATGAGAGGTTTTCC R:TGGTCGTGTAGTGGGGTCC |
| Sequence-based reagent | Etv4 | | PrimerBank ID: 118918427 c2 | F:CAGTGCCTTTACTCCAGTGCC R:CTCAGGAAATTCCGTTGCTCT |
| Sequence-based reagent | Fos | | PrimerBank ID: 254750707 c2 | F:GGGGCAAGGTGGAACAGTTAT R:CCGCTTGGAGTGTATCAGTCA |
| Sequence-based reagent | Dab2 | | PrimerBank ID: 349585059 c1 | F:GTAGAAACAAGTGCAACCAATGG R:GCCTTTGAACCTTGCTAAGAGA |
| Sequence-based reagent | Pdgfa | | PrimerBank ID: 197333758 c1 | F:GCAAGACCAGGACGGTCATTT R:GGCACTTGACACTGCTCGT |
| Sequence-based reagent | Runx1 | | PrimerBank ID: 169790826 c1 | F:CTGCCCATCGCTTTCAAGGT R:GCCGAGTAGTTTTCATCATTGCC |
| Sequence-based reagent | Hprt1 | | PrimerBank ID: 164518913 c1 | F:CCTGGCGTCGTGATTAGTG R:AGACGTTCAGTCCTGTCCATAA |

## NRVM isolation and 2D culture

All animal procedures were performed in compliance with the Institutional Animal Care and Use Committee at Duke University and the NIH Guide for the Care and Use of Laboratory Animals. NRVMs were isolated as previously described (*Jackman et al., 2018*; *Jackman et al., 2016*; *Li et al., 2017*). Briefly, ventricles were harvested from P2 male and female Sprague-Dawley rat pups, minced finely, and pooled before overnight trypsin incubation at 4°C. The following day, the minced ventricular tissue was subjected to several collagenase digestions and filtering steps to yield single-cell suspension. Cells were pre-plated for 1 hr to remove non-myocytes and enrich the NRVM population. The non-adherent cells were resuspended in 2D cardiac medium (DMEM, 10% fetal bovine serum [FBS], penicillin (5 U/ml), and vitamin B12 (2 µg/ml)) and plated onto fibronectin-coated Aclar coverslips at a

density of 5× 10$^5$ cells per well of a 12-well plate. Twenty-four hours following plating, the medium was changed to only include  5% FBS and full media changes were performed every other day.

## Cardiobundle fabrication and 3D culture

NRVM cardiobundles were prepared as previously described (*Helfer and Bursac, 2020*; *Jackman et al., 2018*; *Jackman et al., 2016*). Briefly, 6.5×10$^5$ freshly isolated NRVMs were mixed with a fibrin-based hydrogel (2.5 mg/ml fibrinogen, 1 U/ml thrombin, and 10% v/v/Matrigel) and cast in PDMS tissue molds with two 2 mm×7 mm troughs and a porous nylon frame. In experiments testing, the effects of increased cell/hydrogel volume, our standard cell number and gel volume were scaled up by 1.4-fold. The molds containing the hydrogel-cell mixture were incubated at  4°C for 45 min to allow the hydrogel to fully polymerize and attach to the nylon frame. Tissues were then immersed in 3D cardiac medium (Low Glucose DMEM,  10% horse serum,  1% chick embryo extract, aminocaproic acid [1 mg/ml], L-ascorbic acid 2-phosphate sesquimagnesium salt hydrate [50 µg/ml], penicillin [5 U/ml], and vitamin B12 [2 µg/ml]). The following day, the cardiobundles on frames were carefully removed from the molds and cultured in free-floating dynamic conditions on a rocker. Full media changes of 2 ml per well were performed every other day for 14 days.

For experiments utilizing small molecule inhibitors of Erk or Mek, we used SCH772984 (Erki) or PD0325901 (Meki). Both inhibitors were applied from days 8 to 14 after tissue generation at 100 nM, and were compared to a DMSO vehicle control. See Materials table for drug information.

## hiPSC maintenance and CM differentiation

BJ fibroblasts from a healthy male newborn (ATCC cell line, CRL-2522) were reprogrammed episo-mally into hiPSCs at the Duke University iPSC Core Facility and named DU11 (Duke University clone #11) following verification of pluripotency as described previously (*Shadrin et al., 2017*). hiPSCs were maintained as feeder-free cultures on ESC-Matrigel in mTeSR Plus medium and colony-passaged as small (10–20 cells) clusters every 3 days using 0.5 mM EDTA (1:10 split ratio). All hiPSC-CM experiments were performed using DU11 hiPSCs between passages 24 and 45. Mycoplasma testing was done routinely with commercially available testing kits (MycoAlert, Lonza).

hiPSCs were differentiated into CMs via small-molecule-based modulation of Wnt signaling, as previously described (*Lian et al., 2012*). Briefly, DU11 hiPSCs were dissociated into single cells using Accutase and plated into ESC-Matrigel coated dishes at 4×10$^5$/cm$^2$ with 5 µM Y27632 (ROCK inhibitor). Maintenance media was changed daily prior to differentiation. To induce cardiac differentiation (day 0), cells were incubated in 6 µM CHIR99021 in RPMI-1640 with B27(−) insulin. Exactly 48 h later, CHIR was removed and replaced with basal RPMI/B27(−) medium. On day 3, the cells were incubated in RB- containing 5 uM IWR-1, which was switched to basal RPMI/B27(−) medium on day 5. From days 7 to 10, the cells were cultured in RPMI/B27(+)-insulin with media changed every 2–3 days. Spontaneous contractions of hiPSC-CMs started on days 7–10 of differentiation. Differentiating CM cultures were purified via metabolic selection between days 10 and 14 (*Tohyama et al., 2013*), by incubation in 'no glucose' medium (glucose-free RPMI supplemented with 4 mM lactate, 0.5 mg/ml recombinant human albumin, and 213 µg/ml L-ascorbic acid 2-phosphate) for 48 h. At the end of the selection period, cultures were dissociated into single cells using  0.05% trypsin/EDTA followed by quenching with stop buffer (DMEM,  20% FBS, 20 µg/ml DNAse I [Millipore 260913]) and replated onto fresh Matrigel-coated dishes to remove dead cells and debris. Cells were then maintained in our 'standard' media consisting of RPMI/B27(+)-insulin with 2 mg/ml aminocaproic acid, 50 µg/ml L-ascorbic acid 2-phosphate, 0.45 µM 1-thioglycerol,  1% pen/strep,  1% non-essential amino acids, and  1% sodium pyruvate. hiPSC-CM cultures with greater than  90% cTnT positive cells as measured by flow cytometry were used for all experiments. For select experiments, we used maturation media (Mills MM) by *Mills et al., 2017* consisting of DMEM with no glucose and no glutamine, with  4% B27 without insulin,  1% GlutaMAX, 200 µM L-ascorbic acid 2-phosphate sesquimagnesium salt hydrate, 1 % penicillin/streptomycin, 1 mM glucose, and 100 µM BSA-palmitate saturated fatty acid complex or control media from the same group (Mills Ctrl) consisting of α-MEM with GlutaMAX,  10% FBS, 200 µM L-ascorbic acid 2-phosphate sesquimagnesium salt hydrate, and  1% Penicillin/Streptomycin.

Experiments utilizing Erk or Mek inhibition were performed using the same small molecules described for NRVM cardiobundles, but we also performed dose-response testing from 10 nM to 1 µM of both compounds and compared to a DMSO vehicle control. Erki or Meki was applied for

24 hr, media was then changed to include new inhibitor and 10 µM EdU for another 24 hr prior to collection for flow cytometry analysis.

## Cloning of mitogen constructs

For generation of mitogen constructs caYap8SA, caCtnnb1, Ccnd2 OE, carErbb2, and cahErbb2, plasmid containing the gene sequence was used as PCR template for amplification of inserts prior to cloning (see Key Resources Table for plasmids). PCR primers were designed to add complementary restriction site overhangs to the gene inserts which were also present in the MHCK7-MCS-P2A-mCherry backbone used for cloning the constructs. Standard restriction cloning was used to insert the gene fragments. Sanger sequencing was performed to ensure maintenance of reading frame and correct sequence. See Source data 4 for maps of generated plasmids.

## Preparation of lentivirus vectors

LVs were prepared as previously described (*Rao et al., 2018*). Briefly, Hek293T cells were cultured in high glucose DMEM containing 10% FBS and 1% Penicillin/Streptomycin. Plasmids (Construct plasmid, Pax2, and VSVG) were purified using midiprep before transfection into Hek293T cells at 65–75% confluence using Jetprime transfection reagent. Medium was changed 16 hr following transfection, and medium containing virus was harvested 3–4 days following initial transfection. Virus was purified by precipitation using 3 volumes medium to 1 volume LentiX-Concentrator at 4°C overnight, then pelleted by centrifugation at 1500×g for 45 min at 4°C. Precipitated virus was aliquoted and stored at –80°C before use.

For experiments investigating the effects of the LVs on hiPSC-CM monolayers, cells were transduced with the LVs 17–20 days following the initiation of differentiation and were maintained for 1 week to allow the LVs to reach maximal expression before terminal analysis. For NRVM monolayer experiments, viral suspension was added at the time of cell plating. For NRVM cardiobundle experiments, LVs were added to the hydrogel-cell mixture at the time of cardiobundle fabrication to yield transduction efficiency between 45% and 80%.

## Flow cytometry

NRVM or hiPSC-CM monolayers were rinsed with phosphate-buffered saline (PBS) then dissociated using 0.05% Trypsin-EDTA at 37°C for 3 min, upon which monolayers were triturated several times to yield a single-cell suspension. Trypsin was quenched with DMEM/F12 containing 20% FBS and 20 µg/ml DNase I. The cell suspension was centrifuged at 300×g for 5 min, then resuspended in % paraformaldehyde (PFA) diluted in PBS. Cells were incubated in PFA for 10 min at room temperature (RT), centrifuged again, then resuspended in PBS containing 5% FBS for storage at 4°C.

Cells were stained for flow cytometry after centrifugation at 300×g for 5 min to remove storage medium. If EdU staining was performed, cells were incubated with the EdU flow cytometry staining cocktail as per the manufacturer's protocol (Thermo Fisher Scientific) and incubated in the dark for 30 min, then washed 2× by addition of PBS followed by centrifugation. Antibody staining was performed after EdU staining. For antibody staining, cells were resuspended in FACS buffer (PBS with 0.5% BSA, 0.1% Triton-X 100, and 0.02% Azide). Primary antibodies including an isotype control were diluted in FACS buffer and incubated for 1 hr on ice. Cells were washed 2× with FACS buffer before addition of secondary antibodies and Hoechst diluted in FACS buffer. Secondary antibodies were incubated for 30 min at RT. Samples were run on a BD Fortessa X-20.

## Immunostaining and imaging

Cell monolayers were fixed with 4% v/v PFA at RT for 15 min, then blocked in antibody buffer (5 w/v donkey serum, 0.1% v/v Triton X-100, in PBS) for 2 hr at RT and incubated with primary antibodies for 2 hr in antibody buffer. Primary antibody sources and dilutions are indicated in the Key Resources Table. The monolayers were washed with PBS before incubation with Alexa Fluor-conjugated secondary antibodies at 1:1000 and Hoechst at 1:200 in antibody buffer for 1 hr. Monolayer samples were mounted using Fluoromount-G Mounting Medium and imaged using an Andor Dragonfly spinning disk confocal microscope.

Engineered cardiobundles were fixed with 2% v/v PFA on a rocking platform at 4°C overnight. For cross-sectional analysis, the fixed tissues were suspended in OCT and flash-frozen in liquid nitrogen

until solidified. The frozen tissue blocks were sectioned using a cryostat (Leica) into 10 μm sections. Cardiobundle cross-sections were blocked in antibody buffer for two hours at RT. Whole bundles for longitudinal images were blocked overnight at 4°C. All samples were incubated with primary antibodies at 4°C overnight in antibody buffer. Primary antibodies were used at the indicated dilutions in the Key Resources Table. Samples were incubated with Alexa Fluor-conjugated secondary antibodies at 1:1000 and Hoechst 1:200 in antibody buffer for 2.5 hr at RT for cross-sections and overnight at 4°C for whole bundles. Cross-sections and un-sectioned whole bundles were mounted with hard-set mounting medium (Antifade Glass) and imaged using an Andor Dragonfly spinning disk confocal microscope.

## qPCR

RNA was extracted using RNeasy Plus Mini Kit according to the manufacturer's instructions (Qiagen). Total RNA was converted to cDNA using iScript cDNA Synthesis Kit (Bio-Rad). Standard qPCR reactions with 5 ng cDNA per reaction were performed with iTaq Universal SYBR Green Supermix (Bio-Rad) in the CFX Connect Real-Time PCR Detection System. All primers used are listed in qPCR Primer Table in Key Resources Table.

## Force measurements

After 14 days of culture, cardiobundle force generation was measured using custom-made force measurement setup consisting of an optical force transducer and linear actuator as previously described (*Jackman et al., 2016*). In 37°C Tyrode's solution, the cardiobundle was pinned to chamber at one end and a PDMS float connected to a linear actuator controlled by Labview software at the other end. Using platinum electrodes, a 90 V biphasic electrical pulse was applied for 5 ms at a 2 Hz rate to induce contractions. The force measurements were performed at the ends of 4% stretch steps lasting 45 s ( 0% stretch being the culture length of 7 mm), until 16% stretch was reached. Maximum twitch amplitude (occurring anywhere between 4% and 12% stretch), passive force-length curves, and parameters of twitch kinetics were derived as previously described using custom Matlab software (*Madden et al., 2015*). For contractile force measurements under mechanically unloaded conditions, tissues were relaxed between 0% and –8 % stretch to length where passive force became 0, upon which electrically induced contractions were recorded at 2 Hz rate.

## Western blot

To isolate total protein from hiPSC-CM samples, cells were rinsed twice with ice-cold PBS before lysis with RIPA buffer containing protease inhibitor cocktail (Sigma-Aldrich P8340) and phosphatase inhibitor cocktail 3 (Sigma-Aldrich P0044). Cells were incubated on ice for 10 min, then lysates were collected and spun down at 10,000×*g* to pellet debris. Supernatants were measured using BCA assay to determine total protein concentration. About 30 μg of samples were run on 4–12% gradient gels with Tris-Glycine-SDS running buffer at 100 V for 1–1.5 hr depending on the size of proteins being separated. Proteins were transferred to 0.45 μm PVDF membranes for Western blot at 4°C at 60 V for 2 hr. Membranes were blocked overnight in 3% BSA in Tris-buffered saline (TBS). Membranes were cut such that multiple size proteins could be blotted from the same membranes. Primary antibodies were diluted in 3% BSA and incubated with membranes overnight at 4°C. Membranes were washed 3× with TBS containing 0.1% Tween-20 (TBS-T) before incubation with HRP-conjugated secondary antibodies. Membranes were washed 3× with TBS-T before incubation in SuperSignal West Pico PLUS Chemiluminescent Substrate for 5 min. Membranes were imaged with a Biorad ChemiDoc using signal accumulation mode for up to 2 min. If the protein of interest was similar in size to the housekeeping gene (GAPDH), membranes were stripped following exposure for 10 min using Restore PLUS Western Blot Stripping Buffer. Membranes were then reblocked and re-probed as indicated above. Antibodies and their dilutions can be found in the Materials and methods section.

## Quantification and statistical analysis

Statistical analysis was performed with GraphPad Prism software. Statistical details can be found in *Supplementary file 1*. Outliers were identified and removed using GraphPad Prism 8.3.0 ROUT method (Q=1%). Normality testing was done using the Shapiro-Wilk test and testing for equal variances was done using the Brown-Forsythe test. If data was not normally distributed, we performed

logarithmic transformations and re-tested for normality and equal variances prior to performing the appropriate statistical test. All experiments were carried out in multiple cell batches. To analyze experiments in which not all mitogen groups were studied simultaneously, we first expressed results relative to Ctrl LV group within each experiment and then compared normalized results among different groups by applying logarithmic transformation followed by the appropriate statistical test.

## Image analysis

Image analysis was performed using custom CellProfiler (*Carpenter et al., 2006*) and FIJI (*Schindelin et al., 2012*) macros. Briefly, CellProfiler was used to determine nuclear number as well as vimentin[+] and EdU[+] cells. To quantify nuclear number and EdU+ nuclei, the identify primary objects function was used with global minimum cross entropy thresholding. To quantify vimentin[+] cells, vimentin signal was smoothed using Gaussian filter, thresholded using the minimum cross-entropy function, holes were removed using the remove holes function, and the watershed function was applied for segmentation. Colocalization analysis using the related objects function between vimentin signal and nuclei as well as EdU signal and nuclei was performed to exclude proliferative fibroblasts from cardiomyocyte EdU quantification. A custom FIJI macro using auto-thresholding methods was used to determine cardiobundle F-actin[+] area and total CSA.

## Acknowledgements

The authors thank A Helfer for numerous discussions regarding the experiments performed in this manuscript, and L Baugh for providing feedback on the initial draft of the manuscript. This study was supported by the National Institutes of Health (NIH) grants U01HL134764, R01HL132389, 5T32HD040372, an 1F31HL156453, the Foundation Leducq grant 15CVD03, and a grant from Duke Translating Duke Health Initiative. The content of the manuscript is solely the responsibility of the authors and does not necessarily represent the official views of the National Institutes of Health.

## Additional information

### Funding

| Funder | Grant reference number | Author |
| --- | --- | --- |
| National Institutes of Health | Research Project Cooperative Agreement | Nenad Bursac |
| National Institutes of Health | Research Project Grant | Nenad Bursac |
| National Institutes of Health | Training Grant | Nicholas Strash Sophia DeLuca |
| Foundation Leducq | Transatlantic Networks of Excellence Program | Nenad Bursac |
| Duke University | Translating Duke Health: Cardiovascular Health Initiative | Nenad Bursac |
| National Institutes of Health | U01HL134764 | Nenad Bursac |
| National Institutes of Health | R01HL132389 | Nenad Bursac |
| National Institutes of Health | 5T32HD040372 | Nicholas Strash Sophia DeLuca |
| Foundation Leducq | 15CVD03 | Nenad Bursac |
| National Institutes of Health | 1F31HL156453 | Nicholas Strash |

| Funder | Grant reference number | Author |
|--------|------------------------|--------|

The funders had no role in study design, data collection and interpretation, or the decision to submit the work for publication.

## Author contributions
Nicholas Strash, Conceptualization, Data curation, Investigation, Methodology, Supervision, Validation, Visualization, Writing - original draft, Writing - review and editing; Sophia DeLuca, Data curation, Investigation, Validation, Writing - original draft, Writing - review and editing; Geovanni L Janer Carattini, Investigation; Soon Chul Heo, Ryne Gorsuch, Conceptualization, Methodology; Nenad Bursac, Conceptualization, Funding acquisition, Methodology, Project administration, Supervision, Visualization, Writing - original draft, Writing - review and editing

## Author ORCIDs
Nicholas Strash (iD) http://orcid.org/0000-0003-1907-0925
Nenad Bursac (iD) http://orcid.org/0000-0002-5688-6061

## Ethics
This study was performed in accordance with the recommendations in the Guide for the Care and Use of Laboratory Animals of the National Institutes of Health. All of the animals were handled according to approved institutional animal care and use committee (IACUC) protocol (#A100-18-04) at Duke University.

## Decision letter and Author response
Decision letter https://doi.org/10.7554/eLife.65512.sa1
Author response https://doi.org/10.7554/eLife.65512.sa2

# Additional files

## Supplementary files
• Supplementary file 1. Statistical information for all figures.
• Transparent reporting form
• Source data 1. Source data used to generate figures.
• Source data 2. Unedited, labeled western blots for all figures.
• Source data 3. Unedited, uncropped western blots for all figures.
• Source data 4. Plasmid maps for constructs generated in this study.

## Data availability
All data generated or analyzed during this study are included in the manuscript and supplemental source data file for all figures.

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
