## [Decision Letter]

**Acceptance summary:**

Using in vitro 2-D and engineered cardiac tissue 3-D models the authors analyzed several mitogenic signaling routes in rodent and human cardiomyocytes. Specifically, the authors identified a constitutively active mutant form of human Erbb2 as the most potent mitogen, and that its expression determines cardiomyocyte sarcomere loss, cell proliferation, as well as cardiac tissue structure and function in an ERK-dependent manner. The findings presented may facilitate development of regenerative heart repair strategies following myocardial infarction and will be of interest to scientists and clinician-scientists working in the field of myocardial regeneration.

**Decision letter after peer review:**

Thank you for submitting your article "Human Erbb2-induced Erk Activity Robustly Stimulates Cycling and Functional Remodeling of Rat and Human Cardiomyocytes" for consideration by *eLife*. Your article has been reviewed by 3 peer reviewers, one of whom is a member of our Board of Reviewing Editors, and the evaluation has been overseen by a Senior Editor. The following individual involved in review of your submission has agreed to reveal their identity: Gilles De Keulenaer (Reviewer #3).

Essential revisions

1) Use of post-mitotic human CMs, such as metabolically "matured" CMs or EHTs from hiPSC-CMs would add value to the work. These can be obtained using metabolic stimuli as shown by Mills et al. 2017 (PNAS, https://doi.org/10.1073/pnas.1707316114).

2) Although cell cycle activity was clearly induced, no proof of increased cytokinesis has been presented. Is the increase in size of cardiac bundles due to increased number of CMs or increased size of CMs? Both can lead to the development of a necrotic centre as seen in the cahErbb2 overexpressing cardio bundle. The ERK pathway is key mediator of most hypertrophic stimuli and its activation induces CM hypertrophy (see for example the review by Bueno and Molkentin 2002, Circ Res, https://doi.org/10.1161/01.RES.0000038488.38975.1A). Also the Erbb2 ligand Neuregulin-1 has been shown to induce CM hypertrophy (Zurek et al. 2020, Circulation, https://doi.org/10.1161/CIRCULATIONAHA.119.044313).

Analysis of hypertrophy on either phenotypic or gene expression level in cardiac bundles is needed to exclude hypertrophy as a mechanism behind the increased CSA. For example cross-sectional area of individual cardiomyocytes by using a cell membrane stain combined with a CM marker – or gene expression of hypertrophic markers.

3) All figures: STDEV should be used instead of SEM. Moreover, violin plots or dot plots should be considered to better visualize the distribution of the data.

4) Some concerns about statistics, which require clarification:

i) The N numbers are huge and according to the Table 1 it is unclear if they all indeed represent independent samples. Were several samples from the same differentiations or cell isolations used? One field in a monolayer (Table 1, definition of n in Figures 1J, 1P, 4C, 4E, 4G) should not be considered to represent n=1, if several images were analysed from the same sample and/or if several technical replicates (samples prepared from the same cell isolation or differentiation and treated similarly) were analysed. Similarly, RNA or 10^4 cells from one monolayer do not correspond to n=1, if several technical replicates were used. Only samples from separate differentiations or cell isolations should be considered as representatives of n and the results from technical replicates should be averaged to form the n=1 data.

ii) Were the data analysed for equal variances before selecting the test (parametric vs. non-parametric)? If not, this should be done and an appropriate test chosen.

iii) Why did the authors carry out multiple t tests instead of using ANOVA, which is more suitable for multiple comparisons? For example, randomized block ANOVA is a very useful test for example for analysing data for which baseline values may vary between cell preparations (e.g. the percentage of Edu+ cells). In particular, the statistics for data in Figure 5 – Supplemental figure 1, panel B should be corrected. Why have the authors not included the control sample in the statistical analysis? Analysis of the percentage of Edu+ cells (not normalised data) using randomized block ANOVA solves the issue of experiment/cell-preparation-dependent variability in basal levels and the zero variance in normalised data. The data can be presented as is (normalised to control), but statistics should be corrected.

5) The authors have to be complimented for the high technical quality of their studies. This reviewer does not ask for new experiments, but suggests to underscore the limitations of this study, especially with regards to the observed effects of Erbb2 overexpression.

i) First, authors indicate that a necrotic core develops in the 3D model (because of extensive proliferation and development of hypoxia in the core); the extent to which the necrotic core contributes to an increased stiffness is undetermined, and should be addressed. Also, authors should discuss whether development of a necrotic core is related to the avascular structure of the 3D model?

ii) Second, the lentiviral overexpression of ERBB2 is not transient, because it cannot be turned off. Ideally, an on-off system is used so that after the proliferation phase, cardiomyocytes are allowed to redifferentiate and restructure their sarcomeres. No one is going to advocate continuous overexpression of ERBB2 in any tissue. An on-off system would allow the cardiomyocytes to redifferentiate resulting in a level playing field for all vectors. This issue should be better discussed.

iii) Third, it is not surprising that cardio bundles with the most actively proliferating cardiomyocytes and the most disorganized sarcomeres are less contractile. Are these unloaded cardiomyocytes operating at slack length which would, beside the proliferating phenotype, be an explanation for their hypo-contractility due to increased restoring forces? This question is important, since then hypocontractility may be, at least to some extent, an in vitro artefact.

6) The authors have provided some functional measurements of the cardiomyocyte bundles, however, additional parameters of cardiomyocyte function warrants investigation to properly establish that the induced proliferation translates into improved function.

---

## [Author Response]

Essential revisions1) Use of post-mitotic human CMs, such as metabolically "matured" CMs or EHTs from hiPSC-CMs would add value to the work. These can be obtained using metabolic stimuli as shown by Mills et al. 2017 (PNAS, https://doi.org/10.1073/pnas.1707316114).

To address this comment, we examined whether lentiviral cahErbb2 expression could induce pro-proliferative effects in hiPSC-CMs cultured in maturation medium by Mills et al. 2017. As shown in Figure 1 —figure supplement 3, we found that maturation media reduced the baseline EdU incorporation rate in CMs by half compared to our standard media as well as control media used by Mills et al. Transduction of cahErbb2 in CMs cultured in maturation media significantly increased EdU incorporation and H3P expression compared to transduction of mCherry (control). These results confirmed that mitogenic effects of cahErbb2 are also robust in metabolically matured hiPSC-CMs. In the main text, revisions made in response to reviewers’ comments are highlighted in red. Additional stylistic edits throughout the text were made to improve readability of the manuscript.

2) Although cell cycle activity was clearly induced, no proof of increased cytokinesis has been presented. Is the increase in size of cardiac bundles due to increased number of CMs or increased size of CMs? Both can lead to the development of a necrotic centre as seen in the cahErbb2 overexpressing cardio bundle. The ERK pathway is key mediator of most hypertrophic stimuli and its activation induces CM hypertrophy (see for example the review by Bueno and Molkentin 2002, Circ Res, https://doi.org/10.1161/01.RES.0000038488.38975.1A). Also the Erbb2 ligand Neuregulin-1 has been shown to induce CM hypertrophy (Zurek et al. 2020, Circulation, https://doi.org/10.1161/CIRCULATIONAHA.119.044313).Analysis of hypertrophy on either phenotypic or gene expression level in cardiac bundles is needed to exclude hypertrophy as a mechanism behind the increased CSA. For example cross-sectional area of individual cardiomyocytes by using a cell membrane stain combined with a CM marker – or gene expression of hypertrophic markers.

We thank reviewers for this insightful comment. While we did not definitively show CM cytokinesis, the polyploid fraction within the hiPSC-CM population was not increased with application of most mitogens (Figure 1D,E), suggesting the cells which incorporated EdU ended up dividing (and re-attaining 2N DNA content) instead of becoming polyploid (having 4N or greater DNA content). In NRVMs the situation is less clear, but we measured the F-actin^+^/vimentin^-^ area divided by number of nuclei within this area in cardio bundle cross-sections as a proxy for CM size that we have used in previous publications (e.g., Jackman et al., 2016, *Biomaterials;* Jackman et al. 2018, *Acta Biomaterialia*). This analysis revealed no change in CM size with mitogen expression (new Figure 3F), suggesting that the increase in cardio bundle size for cahErbb2 group was likely caused by an increase in CM numbers rather than the CM size. We also included measurements of relative cell size based on forward scatter data collected in our flow cytometry experiments in hiPSC-CMs and NRVMs (new Figure 1 —figure supplement 1D-E) which suggested that cahErbb2 does not affect cell size in the monolayer setting.

3) All figures: STDEV should be used instead of SEM. Moreover, violin plots or dot plots should be considered to better visualize the distribution of the data.

We converted any remaining bar/line plots to display STDEV rather than SEM, and most bar graphs were converted to box-and-whisker plots to visualize distribution of the data points.

4) Some concerns about statistics, which require clarification:i) The N numbers are huge and according to the Table 1 it is unclear if they all indeed represent independent samples. Were several samples from the same differentiations or cell isolations used? One field in a monolayer (Table 1, definition of n in Figures 1J, 1P, 4C, 4E, 4G) should not be considered to represent n=1, if several images were analysed from the same sample and/or if several technical replicates (samples prepared from the same cell isolation or differentiation and treated similarly) were analysed. Similarly, RNA or 10^4 cells from one monolayer do not correspond to n=1, if several technical replicates were used. Only samples from separate differentiations or cell isolations should be considered as representatives of n and the results from technical replicates should be averaged to form the n=1 data.

We very much appreciate this comment. We have corrected our mistakes regarding using n=1 to represent images from the same sample and instead have averaged images to generate n=1 per biological replicate; the corrected n values are updated in the supplemental statistics table.

We also wanted to clarify that for the rest of our studies in both monolayer and cardio bundle experiments, we did not use technical replicates as individual n’s. Experiments were performed in multiple cell batches/differentiations to ensure reproducibility. Within a cell batch/differentiation, we used n=3 monolayer culture wells or tissues that were independently treated with LVs or small molecules and independently assessed thereafter, not n=1 well or tissue that was treated, collected, and assessed multiple times to generate n=3.

ii) Were the data analysed for equal variances before selecting the test (parametric vs. non-parametric)? If not, this should be done and an appropriate test chosen.

We have corrected our analyses to test for equal variances before selecting the style of ANOVA and, where appropriate, we have used Welch’s ANOVA or Welch’s t test for statistical analysis to account for unequal variances. This information is updated in our new statistics table.

iii) Why did the authors carry out multiple t tests instead of using ANOVA, which is more suitable for multiple comparisons? For example, randomized block ANOVA is a very useful test for example for analysing data for which baseline values may vary between cell preparations (e.g. the percentage of Edu+ cells). In particular, the statistics for data in Figure 5 – Supplemental figure 1, panel B should be corrected. Why have the authors not included the control sample in the statistical analysis? Analysis of the percentage of Edu+ cells (not normalised data) using randomized block ANOVA solves the issue of experiment/cell-preparation-dependent variability in basal levels and the zero variance in normalised data. The data can be presented as is (normalised to control), but statistics should be corrected.

We appreciate this comment and have now applied ANOVA instead of multiple t-tests for our qPCR and western blot data analyses. We omitted to include that our multiple t-tests had included p value correction for multiple comparison, so revising the statistical tests did not affect our conclusions. The corrected statistical test information can be now found in our updated statistics table; we ran ANOVAs to ensure proper correction for multiple comparisons as suggested. We also corrected our figure to include the control sample in the graph for Figure 5 – supplemental figure 1 as suggested.

Through our analysis of the normalized data, we found they achieved a normal distribution following logarithmic transformation and therefore we determined that analysis of the normalized data was appropriate for our statistical analysis. We then applied Brown-Forsythe tests and if variance was found to be significantly different between groups, we used Welch’s ANOVA with Dunnett T3 correction for multiple comparisons rather than the standard ANOVA method. This information is included in our updated statistics table and statistical analysis section in the methods.

5) The authors have to be complimented for the high technical quality of their studies. This reviewer does not ask for new experiments, but suggests to underscore the limitations of this study, especially with regards to the observed effects of Erbb2 overexpression.i) First, authors indicate that a necrotic core develops in the 3D model (because of extensive proliferation and development of hypoxia in the core); the extent to which the necrotic core contributes to an increased stiffness is undetermined, and should be addressed.

To address this comment, we generated cardio bundles using increased hydrogel volume and cell number, which resulted in a larger tissue size and formation of acellular core (new Figure 3 —figure supplement 3A-F) like those observed in cahErbb2-expressing cardio bundles. However, this increase in tissue size and core formation did not significantly alter passive force of the cardio bundles, in contrast to cahErbb2 expression that yielded 8-fold higher passive forces (Figure 3H). This suggested to us that cahErbb2-expressing cardio bundles have increased stiffness not due to the presence of acellular core, but due to an increase in ECM stiffness (e.g., accumulated collagen I), cell stiffness (e.g., more abundant intermediate filaments), or both. We thus performed additional quantitative immunostaining analyses in cahErbb2-expressing cardio bundles and found that collagen I abundance was not changed (new Figure 3 —figure supplement 3G), while abundances of intermediate filament proteins nestin and vimentin were significantly increased (new Figure 3 —figure supplement 3I-K). Collectively, these results suggested that increased cell stiffness may be the main contributor to increased tissue stiffness induced by cahErbb2 expression.

Also, authors should discuss whether development of a necrotic core is related to the avascular structure of the 3D model?

Owing to the lack of functional vasculature, cell survival in cardio bundles depends on the adequate diffusion of oxygen and nutrients from surrounding culture media into the center of the tissue. Thus, the mitogen-induced increase in cardio bundle size, along with a potentially higher metabolism of proliferating CMs, likely caused reduced levels of oxygen/nutrients in the cardio bundle center leading to the formation of the necrotic core. In the revised manuscript, we now discuss roles of the avascular nature of cardio bundles in development of necrotic core.

ii) Second, the lentiviral overexpression of ERBB2 is not transient, because it cannot be turned off. Ideally, an on-off system is used so that after the proliferation phase, cardiomyocytes are allowed to redifferentiate and restructure their sarcomeres. No one is going to advocate continuous overexpression of ERBB2 in any tissue. An on-off system would allow the cardiomyocytes to redifferentiate resulting in a level playing field for all vectors. This issue should be better discussed.

We agree with reviewers that constitutive expression of caErbb2 or any other mitogens would not be a safe approach to cardiac therapy. In our studies, we constitutively expressed mitogens to facilitate direct comparison of how they affect CM structure and function. Some potential effects of the cahErbb2 turn-off after 7-day expression could be inferred from our Meki and Erki experiments. Per reviewers’ suggestion, in the revised manuscript, we additionally discuss effects of continuous vs. transient mitogen expression in CMs.

iii) Third, it is not surprising that cardio bundles with the most actively proliferating cardiomyocytes and the most disorganized sarcomeres are less contractile. Are these unloaded cardiomyocytes operating at slack length which would, beside the proliferating phenotype, be an explanation for their hypo-contractility due to increased restoring forces? This question is important, since then hypocontractility may be, at least to some extent, an in vitro artefact.

We thank the reviewer for this insightful comment. We measure active (contractile) and passive forces in cardio bundles at multiple tissue lengths, i.e., starting from the culture length of 7mm, then increasing the length in 4% steps and measuring active and passive force at each step after the tissue equilibrates. This allows us to both find the maximum active force that tissue generates and construct passive force-length curves. Generally, with this protocol, our control cardio bundles produced maximum active forces at 8-12% stretch, while cahErbb2expressing cardio bundles produced maximum active force at ~4% stretch. These maximum active forces were presented and compared among groups. To address the reviewers’ comment, we performed additional experiments shown in new Figure 3 – supplemental figure 2 (panels D, E), where we decreased tissue length (negative %stretch) to a point when passive tension became 0 (~slack length). At these equally unloaded conditions, we compared generated active forces and confirmed that cahErbb2-expressing tissues still produced significantly lower forces compared to control tissues.

6) The authors have provided some functional measurements of the cardiomyocyte bundles, however, additional parameters of cardiomyocyte function warrants investigation to properly establish that the induced proliferation translates into improved function.

We note that sustained cahErbb2 expression in cardio bundles induced deterioration rather than improvement of contractile function. To address reviewers’ request for measurement of additional functional parameters, we included analyses of the twitch kinetics in our tissues (new Figure 3 —figure supplement 2A-C and Figure 5 —figure supplement 2). These measurements serve as additional readouts of CM function and together indicate that the increased Erk activity because of cahErbb2 activation slows kinetics of force generation in cardio bundles. As noted in our answer to Reviewers’ comment 5 (ii), transient rather than continuous upregulation of mitogen expression would offer a possibility to induce short-lasting proliferation of CMs, followed by their redifferentiation and improvement of cardiac function, as shown previously by D’Uva et al., (Nat Cell Biol., 2015).